# Histopathological Chromogranin A-Positivity Is Associated with Right-Sided Colorectal Cancers and Worse Prognosis

**DOI:** 10.3390/cancers13010067

**Published:** 2020-12-29

**Authors:** Zoltan Herold, Magdolna Dank, Magdolna Herold, Peter Nagy, Klara Rosta, Aniko Somogyi

**Affiliations:** 1Department of Internal Medicine and Oncology, Semmelweis University, Tomo u. 25-29., H-1083 Budapest, Hungary; dank.magdolna@med.semmelweis-univ.hu; 2Department of Internal Medicine and Hematology, Semmelweis University, Szentkiralyi u. 46., H-1088 Budapest, Hungary; herold.magdolna@med.semmelweis-univ.hu (M.H.); somogyi.aniko@med.semmelweis-univ.hu (A.S.); 31st Department of Pathology and Experimental Cancer Research, Semmelweis University, Ulloi ut 26., H-1085 Budapest, Hungary; nagy.peter@med.semmelweis-univ.hu; 4Department of Obstetrics and Gynecology, Medical University of Vienna, Wahringer Gurtel 18-20, A-1090 Vienna, Austria; klara.rosta@meduniwien.ac.at

**Keywords:** chromogranin A, chromogranin B, colorectal neoplasms, interleukin-6, neuroendocrine cells, survival analysis, thrombocytosis, thrombopoietin

## Abstract

**Simple Summary:**

Several factors are known to affect colorectal cancer (CRC) patient survival, including elevated platelet counts (thrombocytosis) and chromogranin A-positive neuroendocrine-cell differentiation (CgA^+^). Thrombocytosis can occur due to biochemical changes caused by the tumor itself (known as paraneoplastic thrombocytosis) or due to the bleeding of the tumor (reactive thrombocytosis). Our effort was primarily focused on (1) determining if CgA^+^ and paraneoplastic thrombocytosis combined can affect CRC and (2) finding out if there is a possible connection between the two. With the help of chromogranin A immunohistochemical staining, the measurement of circulating biochemical markers of paraneoplastic thrombocytosis (interleukin-6 and thrombopoietin) and chromogranins A and -B, indication was found that CRC combined with CgA^+^ has a well-distinguishable pathophysiology, compared to CRCs without CgA^+^. A possible, new subtype of CRC is proposed, which can be identified easily with chromogranin A immunohistochemical staining. However, its impact should be further studied.

**Abstract:**

Background: Colorectal cancer (CRC) is known to be affected by paraneoplastic thrombocytosis and chromogranin A-positive neuroendocrine-cell differentiation (CgA^+^). Their combined effect has never been previously investigated. Methods: A prospective cohort pilot study of 42 CRC patients and 42 age- and sex-matched controls was carried out. Plasma interleukin-6, thrombopoietin, and serum chromogranin A and -B were measured; furthermore, tumor tissue was immunohistochemically stained for CgA^+^. Results: Twenty-seven and 15 patients were assigned to the chromogranin A-negative (CgA^−^) and CgA^+^ groups, respectively. Within the CgA^+^ group, right-sided tumors were more frequent (18.5% vs. 53.3%), no stage I cancer was found, and patients of this group were in worse general condition. Compared to control subjects, chromogranin A level was higher in the CgA^+^ group (*p* = 0.0086), thrombopoietin (*p* = 0.0040) and chromogranin B (*p* = 0.0070) in the CgA^−^ group, while interleukin-6 was high in both tumor groups (*p* ≤ 0.0090). Survival was significantly worse in the CgA^+^ group (hazard ratio: 5.73; *p* = 0.0378). Conclusions: Different thrombopoietin levels indicated distinct thrombocytosis types. Within the two CRC groups, serum levels of chromogranins changed in different directions suggesting two well-distinguishable pathophysiologies. Based on these observations we propose a new subtype of CRC, which can be characterized by chromogranin A-positive neuroendocrine-cell differentiation.

## 1. Introduction

Colorectal cancer (CRC) is one of the most common cancers; in 2018 over a million new cases were registered in 185 countries with more than 550,000 deaths [1]. Based on the data available from the National Cancer Registry of Hungary, over 10,000 new CRC cases were registered in 2017, of which nearly 56% were male and 44% female and the disease developed more often in elderly persons. Moreover, the number of newly registered cases is slowly rising every year, in line with international trends [1,2].

Chromogranins A (CgA) and B (CgB) are produced by neurons and endocrine and neuroendocrine cells throughout the human body [3]. Besides the role of CgA as one of the primary biochemical marker in routine neuroendocrine diagnostics [4,5], it was proposed recently as a promising biomarker for early diagnosis of CRC [6,7]. CgB was suggested as a substitute to CgA when the measurement of CgA is limited [8,9,10].

A number of prognostic biomarkers, which worsen the survival of patients have been validated in CRC. These include high platelet counts (thrombocytosis) [11] and chromogranin A-positive neuroendocrine cell (CgA^+^) differentiation within the colorectal adenocarcinoma [12,13,14,15]. Neither the exact background mechanisms of the latter nor the former are sufficiently clarified to date; nor do we have any available information on a possible relationship between the two. Therefore, a prospective cohort pilot study was carried out to search for relationships between CRC and the combined effect of thrombocytosis and CgA^+^ differentiation.

## 2. Results

Forty-two CRC patients were included in the study, and patients were divided into two cohorts based on CgA-positive immunostaining. Patients having no neuroendocrine-cell differentiation were assigned to the CgA^−^ group (*n* = 27), whereas patients with positive results to neuroendocrine-cell differentiation were assigned to the CgA^+^ group (*n* = 15). A scattered pattern of the CgA^+^ cells was observed within 10 CgA^+^ patients, while in the remaining 5 patients a grouped pattern could have been found. Furthermore, 42 propensity-score-matched control subjects were also enrolled to the study for the comparison of non-conventional parameters (CgA, CgB, interleukin-6, and thrombopoietin).

### 2.1. Preoperative Measurements

There was no statistical difference with respect to sex (*p* = 1.0000), age (*p* = 0.9113), comorbidities (*p* ≥ 0.3831), chemotherapy (*p* = 1.0000), or the usage of biological therapies (*p* = 1.0000) within the two CRC cohorts. The low frequency of biological therapy within the CgA^+^ group was due to worse general condition of patients. Right-sided tumor occurred tendentiously less often (18.5%) in the CgA^−^ group than within the CgA^+^ group (53.3%, *p* = 0.2462; without false discovery rate (FDR) correction *p* = 0.0352). No difference was found within staging of the tumors (*p* = 0.3831), but it has to be highlighted that stage I cancer was observed only in the CgA^−^ group (Table 1). Stages II, III, and IV occurred in six (scattered), two (grouped) and seven (four scattered and two grouped) CgA^+^ patients, respectively.

Body mass index (*p* = 0.0465) and waist (*p* = 0.0099) and hip circumference (*p* = 0.0222) were significantly lower within CgA^+^ patients (Table 2). Weight loss over 10 kg within a year was observed in five and two cases in the CgA^+^ and CgA^−^ groups, respectively. Lymphocyte count (*p* = 0.0426), hemoglobin (*p* = 0.0099), hematocrit (*p* = 0.0119), and mean corpuscular hemoglobin concentration (*p* = 0.0426) was significantly lower in the CgA^+^ group. Furthermore, tendentiously lower values of mean corpuscular hemoglobin (*p* = 0.1212; without FDR *p* = 0.0582) and higher values of red blood cell distribution width (*p* = 0.1186; without FDR *p* = 0.0522) were observed. The observed frequency of hematochezia (*p* = 0.1804) and fecal occult blood test positivity (*p* = 0.6979) did not differ between the two tumor groups. Despite that there was no difference between the number of major cardiovascular events prior to CRC (*p* = 0.3831) or in the usage of statins (*p* = 0.3831) and agents acting on the renin–angiotensin system (*p* = 0.8947; Table 1), low-density lipoprotein cholesterol (LDL-C) levels were significantly lower in the CgA^+^ group (*p* = 0.0119). Serum total protein (*p* = 0.0256) and serum albumin (*p* = 0.0222) was also significantly lower in CgA^+^ patients (Table 2). Carcinoembryonic antigen and carbohydrate antigen 19-9 levels were tendentiously lower in the CgA^−^ group (Figure A1). Furthermore, carcinoembryonic antigen positively correlated with serum CgA levels (Spearman’s rho = +0.38; *p* = 0.0162), while with carbohydrate antigen 19-9 marginal association was found (Spearman’s rho = +0.29, *p* = 0.0737).

### 2.2. Measurement of Chromogranins, Interleukin-6 and Thrombopoietin

Compared to control participants, serum CgA was not different in the CgA^−^ group (*p* = 0.2331), but was significantly higher in the CgA^+^ group (*p* = 0.0086). The two tumor group differed marginally (*p* = 0.0908; Figure 1A). CgA is heavily affected by antacid usage [8], and ~31% of the CRC patients used some kind of antacid (Table 1). Therefore, we have analyzed our data excluding those patients as well, and a trend similar to that in all patients was observed—higher values could be found within the CgA^+^ group. However, due to the even lower number of cases, the power of the statistical test decreased further (CgA^−^: 50.45 ± 20.06 ng/mL; CgA^+^: 64.00 ± 35.64 ng/mL; *p* = 0.1024).

CgB was significantly higher in the CgA^−^ group (*p* = 0.0070), compared to the control subjects, while the CgB level of CgA^+^ patients did not differ from the level in either the control (*p* = 0.8037) or the CgA^−^ groups (*p* = 0.1014; Figure 1B). Similar to CgB, the thrombopoietin level was only significantly different in the CgA^−^ group (*p* = 0.0040; Figure 1D). Plasma interleukin-6 levels of both CRC groups was significantly higher, compared to those in the control group (CgA^−^: *p* = 0.0037, CgA^+^: *p* = 0.0090), but did not differ between the tumor cohorts (*p* = 0.9483; Figure 1C).

Correlation between chromogranins and inflammatory parameters (high sensitivity C reactive protein, interleukin-6, and white blood cell count) showed that CgA was not correlated with these parameters in any of the tumor groups (*p* ≥ 0.0824). In the CgA^−^ group, CgB was correlated to high sensitivity C reactive protein (Spearman’s rho = +0.54; *p* = 0.0088) and interleukin-6 (Spearman’s rho = +0.55; *p* = 0.0088); while no correlation was found within CgA^+^ patients (*p* ≥ 0.8544).

### 2.3. Analysis of Conventional and Personalized Indicator Thrombocytosis (PIT)-Based Thrombocytosis

Conventional thrombocytosis (platelet count > 400 × 10^9^/L) was observed in two and four cases within the CgA^−^ and CgA^+^ groups, respectively (*p* = 0.1642). Individualized platelet changes (PIT thrombocytosis) can better represent the general condition of patients [17], but to calculate PIT values, a complete blood count result, at least 2.5–5 years prior to the diagnosis of CRC is necessary. Twenty-three of the 42 CRC patients (55%) had these laboratory results, 15 and 8 within the CgA^−^ and CgA^+^ groups, respectively. Based on our pilot analysis, PIT values and their anemia-corrected variants appeared to be higher in the CgA^+^ group (Figure A2) even though aggregation inhibition therapy was more common in the CgA^+^ group (CgA^−^: 2 of the 15 patients (13%), CgA^+^: 5 of the 8 patients (62.5%), *p* = 0.0257).

### 2.4. Postoperative Measurements

To determine whether the various laboratory differences still exist between the two tumor cohorts after the surgical removal of the primary tumor, patients were recalled for a follow-up measurement. We were able to reach 31 patients (call-back rate 74%). The mean duration after tumor resection was 55 ± 12 days. Compared to the preoperative laboratory results, the difference in body mass index (*p* = 0.0048), waist (*p* = 0.0011) and hip circumference (*p* = 0.0085) was still observable between the two study groups. The lower values observed in complete blood count result persisted after surgery as well, while normalization of lymphocyte count, serum albumin, total protein, and LDL-C was observed in the CgA^+^ group (Table 3).

A marginally significant decrease in plasma interleukin-6 levels was observed in the CgA^−^ group (*p* = 0.0995), while there was no change in the CgA^+^ group (*p* = 0.6398). Thrombopoietin levels remained the same in both groups (CgA^−^: *p* = 0.2163; CgA^+^: *p* = 0.8871; Table 4). Follow-up measurement of CgA could not be adequately evaluated due to the low number of cases and the high incidence of antacid usage.

### 2.5. Survival Analysis

For the competing risk survival models, two endpoint events had been defined: (1) death related to colorectal cancer and (2) death related to postoperative complications. The first event occurred in seven (14.3%) and the latter in three (7.1%) cases; surviving patients were followed-up no later than 31 May 2020. CgA-positive neuroendocrine-cell differentiation of the tumor introduced a 5.73-times higher risk for shorter survival (95% confidence interval (CI): 1.10–29.82; *p* = 0.0378; Figure 2). Adjusted survival analyses, stratified by sidedness and staging were performed as well. Assuming different baseline hazards for patients with left- and right-sided CRCs, patients of the CgA^+^ group had a significant (6.99-times) higher risk for shorter survival (95% CI: 1.35–36.18; *p* = 0.0204). Stratification by staging showed no difference between the two groups (*p* = 0.1630).

The hazard ratios for preoperative platelet count has been calculated as well. Each 10 unit increase in platelet count was associated with a hazard ratio of 1.14 (95% CI: 1.05–1.24; *p* = 0.0029), in line with previous observations [11].

## 3. Discussion

The presence of several factors in CRC are known to be poor prognostic signs, these include thrombocytosis and neuroendocrine-cell differentiation of the tumor [11,13,14,18,19]. Both preoperative and postoperative thrombocytosis affects patient survival [11,20]; and in addition to bleeding of the tumor, a metabolic change caused by the tumor, called paraneoplastic thrombocytosis was also suggested [11,20,21]. In paraneoplastic thrombocytosis, higher interleukin-6 production of the tumor modulates thrombopoietin production of the liver, which ultimately causes a significant increase in bone marrow platelet production [21]. In this study, we measured higher plasma interleukin-6 and thrombopoietin levels in the CgA^−^ group, supporting the occurrence and role of paraneoplastic thrombocytosis. However, in the CgA^+^ group only interleukin-6 was significantly higher and thrombopoietin levels were similar to those in age- and sex-matched control subjects. Furthermore, both “traditional”- and PIT-based thrombocytosis [17] was more common in the CgA^+^ group, despite the fact that these patients received aggregation inhibition therapy more frequently. These biochemical differences between the two tumor groups suggest that the cause behind platelet elevation is not the same in those groups, similar to that observed in the case of right and left-sided CRCs [22,23]—in all probability. In the CgA^−^ group paraneoplastic-, while in the CgA^+^ group reactive thrombocytosis may occur.

There are well known differences between right- and left-sided CRCs—cancers of the right colon are predominant in women; their incidence is lower than left-sided ones but is constantly increasing, and they are typically recognized at more severe clinicopathological stages, resulting ultimately in worse overall survival [22,24,25,26]. Similar to the findings of the current results, a previous study also reported that CgA-positive cases occur more often in right-sided CRCs [15]. It must be stated, that compared to our, and the previously mentioned results [15], most neuroendocrine tumors will develop in the cecum and rectum [27]. In addition to the above, hematochezia is more common in left-sided CRCs, while iron deficiency anemia, secondary to occult blood loss occurs more frequently in right-sided CRCs [22,23,24]. Thrombocytosis can occur at both sides of the colon, with no difference to sidedness [28]. Here, we could extend the previous findings with the following: biochemical markers of thrombocytosis, namely circulating interleukin-6 and thrombopoietin level of CRC patients were different if the tumor could have been characterized with the occurrence of CgA-positive differentiated cells; both marker appeared to be higher in the CgA^−^ cohort. CgA^+^ was more common in right-sided CRCs.

Approximately 1% of the epithelial cells in the large intestines comprise endocrine cells [19], which are rich in chromogranins [3,29,30,31]. Neuroendocrine-cell differentiation, including CgA^+^ differentiation, is known to occur more often in right-sided CRCs and in the tumors of the rectum [15]. Compared to neuroendocrine-cell-negative tumors, CgA^+^ neoplasms have been reported to have worse clinical and pathological properties; and we learn more and more about the role of CgA in non-endocrine tumors every day. For example, the anti-tumor potency of the C-terminal CgA_410–439_ cleavage product was described recently [32], but the role of CgA in routine examination and treatment is still unclear [19]. In the current study, we could also report that CRC patients with CgA^+^ differentiation can be characterized with tumors occurring more frequently on the right-sided colon. Furthermore, in the CgA^+^ group only tumors with stage II or higher were found, with worse pre- and postoperative clinical parameters. The only difference between the two study groups, in which the observed results were clinically worse within the CgA^−^ group, was that LDL-C, which is a known risk factor in several tumors, was significantly higher [33,34]. The reason that in the CgA^+^ group, LDL-C was lower, almost within normal range, is unclear. Both CgA and CgB was also reported to be increased in conditions that cause or are associated with inflammation [35,36,37]. We found that CgA levels of CRC patients was not associated with any inflammatory markers, however, CgB was positively correlated with interleukin-6 and high sensitivity C reactive protein, supporting the theory that CgA^+^ and CgA^−^ tumors have different molecular mechanisms, including that the latter are more probably associated with paraneoplastic inflammatory mechanisms, in which interleukin-6 is overproduced.

Despite the rising number of studies reporting a large number of CgA^+^ differentiated colorectal adenocarcinomas [13,14,15,18,19] CgA^+^ differentiation is basically not used in its routine diagnostics. Here, we reported that serum CgA levels are higher in CRC patients with CgA^+^ differentiation, than in the control and CgA^−^ groups. Furthermore, serum CgB measurements showed that within the CgA^+^ group serum CgB levels were the same as those in control subjects, while in CgA^−^ they were significantly higher. The observation that serum CgA was higher and CgB did not change in the CgA^+^ group is similar to that demonstrated in a carcinoid of the colon [38]. The differences in serum CgA and CgB levels and that the two changed to different extents suggests a different expression pattern of the two chromogranins between the tumors affected and those that are not affected by CgA^+^ differentiation. Based on the results of the various biochemical markers within the two study groups we hypothesize a possible new CRC subtype, which is further strengthened by the fact that the mechanism of thrombocytosis also differs between the two groups.

### Limitations of the Study

The main limitation of the current pilot study was the low sample size, which allowed us to demonstrate only different trends in the case of a few parameters (for example PIT variants) between the two tumor groups. Commercially available detection kits of CgA were reported to have different specificity and sensitivity [39], and the usage of only one of those (serum samples: CISbio CGA-RIACT; immunohistochemistry: Dako DAK-A3) might also introduce some bias. Subgrouping the two tumor groups, for example to investigate the effect of staging/differentiation of the tumors/metastasis was not possible due to the low number of cases. Furthermore, the high incidence of antacid introduction after tumor diagnosis prevented us from comparing postoperative CgA level of patients with preoperative measurements, therefore, further studies are recommended to clarify these questions.

## 4. Materials and Methods

The study was conducted in accordance with the WMA Declaration of Helsinki; handling of patient data was in accordance with the General Data Protection Regulation issued by the European Union. The study was approved by the Regional and Institutional Committee of Science and Research Ethics, Semmelweis University (SE TUKEB 21-13/1994, approval date of latest modification: 15 January 2019) and by the Committee of Science and Research Ethics, Hungarian Medical Research Council (ETT TUKEB 8951-3/2015/EKU).

### 4.1. Patients and Study Design

A total of 42 patients diagnosed with CRC and 42 age- and sex-matched control subjects, who attended at the Department of Internal Medicine and Hematology, Semmelweis University, Budapest, from 2016 to 2019, were enrolled for the study. Written informed consent was collected from all study participants before any study-specific procedures. Exclusion criteria for both populations included age < 18 years, any previous malignancies, known hematologic- and/or inflammatory bowel- and/or systemic autoimmune- and/or chronic kidney- and/or inadequately controlled thyroid diseases, the usage of erythropoiesis-stimulating agents and/or recent blood transfusion, and patients with an Eastern Cooperative Oncology Group (ECOG) Performance Status > 2. In addition to the exclusion criteria above, control subjects were not allowed to use any antacids (including proton pump inhibitors and histamine antagonists) or have any conditions that are known to cause changes in serum CgA levels [8]. Due to patient safety, antacid therapy was never suspended in CRC patients.

A prospective cohort pilot study was carried out, where cohorts within CRC patients were defined by CgA-positive immunostaining of the tumor tissue. Age- and sex-matched control subjects were selected via propensity score matching from a total of 157, non-CRC voluntary control participants.

### 4.2. Clinical and Laboratory Data Measurements

Anamnestic, bodyweight, height, waist- and hip circumference data were collected, and fasting blood samples were drawn (1) at CRC diagnosis prior to any oncological treatments or the surgical removal of the primary tumor, and (2) at least six weeks after surgery. Complete blood count, lactate dehydrogenase, high- and low-density lipoprotein cholesterols, creatinine, high-sensitivity C-reactive protein, serum total protein, and albumin was measured at the Central Laboratory of Semmelweis University. Estimated glomerular filtration rate was calculated using the Chronic Kidney Disease-Epidemiology Collaboration equations [40]. Personalized indicator thrombocytosis (PIT) values were calculated as described previously [17].

In addition to routine laboratory, plasma interleukin-6 and thrombopoietin, and serum CgA and CgB levels were measured using the ELECSYS^®^ interleukin-6 ECLIA (Roche Diagnostics GmbH, Mannheim, Germany), the Human Thrombopoietin Quantikine^®^ ELISA (catalog number: DTP00B, R&D Systems, Minneapolis, MN, USA), the CGA-RIACT radioimmunoassay (CISbio International, Gif-sur-Yvette, France), and the Human Chromogranin B (CHGB) ELISA (dilution 10:1, abx151068, Abbexa Ltd., Cambridge, UK) kits, respectively. Chromogranins are known to cleave to smaller peptide fractions [41], as per manufacturer descriptions, both chromogranin kits detect only larger protein sections.

Staging was given by histopathological examination of surgical specimens; the American Joint Committee on Cancer grouping was used [16]. Sidedness of tumor was described as right-sided colon if the tumor was originating from the cecum, ascending colon, or proximal two-third of the transverse colon; and left-sided if originating from the distal one-third of the transverse colon, descending colon, sigmoid colon, or rectum [42]. Chemotherapy was grouped as adjuvant if no metastasis and first- second-, or third-line if metastasis was present. Usage of biological agents was also indicated in the metastatic group.

### 4.3. Chromogranin A-Specific Histopathological Analysis

CgA-specific immunohistochemical staining of resected tumor segments was carried out with the Clone DAK-A3 Monoclonal Mouse Anti-Human Chromogranin A kit (M0869, dilution 1:1000; Dako Denmark A/S, Glostrup, Denmark). CgA-positivity was defined as previously described by Gulubova and Vlaykova [12]. Adenocarcinoma samples, in which anti-CgA-stained cells infiltrated beyond the muscularis mucosae were defined as CgA-positive. Two well-distinguishable patterns were observed: (1) CgA-positive cells were located randomly and individually or (2) the presence of a grouped pattern was observed in the remaining samples (Figure 3). Samples were blind-examined—the pathologist performing the immunohistochemistry was unaware of clinical and survival data of patients.

### 4.4. Statistical Analysis

Statistical analysis was performed with R version 4.0.2 [43] with the R-packages *broman*, *brunnermunzel*, *Matching*, *mstate* and *survival*. The nonparametric Brunner−Munzel test, Fisher’s exact test, Spearman’s rank correlation, permutation-based paired *T*-test, Kruskal−Wallis test with false discovery rate (FDR) corrected pairwise Brunner−Munzel tests as post-hoc, propensity score matching, and cause-specific competing risk survival models were used. Results were expressed as median ± standard deviation and as the number of observations; *p* < 0.05 was considered as statistically significant and *p*-values were corrected with the FDR method for multiple comparisons problem.

Boxplots were drawn with the defaults of R [43] −the box represents the interquartile range (IQR), the solid line within the box represents the median, the upper whisker is defined as the smaller value from the maximum or upper quartile +1.5 IQR, and the lower whisker is the larger value from the minimum or lower quartile −1.5 IQR.

## 5. Conclusions

Summarizing our pilot study, according to our results, patients with colorectal adenocarcinoma differentiated by CgA^+^ cells had worse survival and clinicopathological features. The different thrombopoietin, CgA, and CgB levels in the CgA^+^ group suggested a difference in the mechanism of thrombocytosis and chromogranin production, compared to that of CgA^−^ CRC patients. The effect of thrombocytosis on CRC patient survival is well known [11,20], however, the underlying mechanism has not yet been adequately clarified, and the results of the present pilot study can make a contribution to a more accurate understanding of these mechanisms. A limitation of the current study was the small sample size, which allowed us to observe only trends in a few parameters. Based on our results we propose a new subtype of colorectal cancer, which can be characterized by CgA^+^ differentiation, therefore we suggest the routine usage of CgA-specific immunohistochemical staining of colorectal tumor specimens. To support these assumed causative relationships and assess the role of treatment, further functional studies and randomized clinical trials are needed.

## Figures and Tables

**Figure 1 cancers-13-00067-f001:**
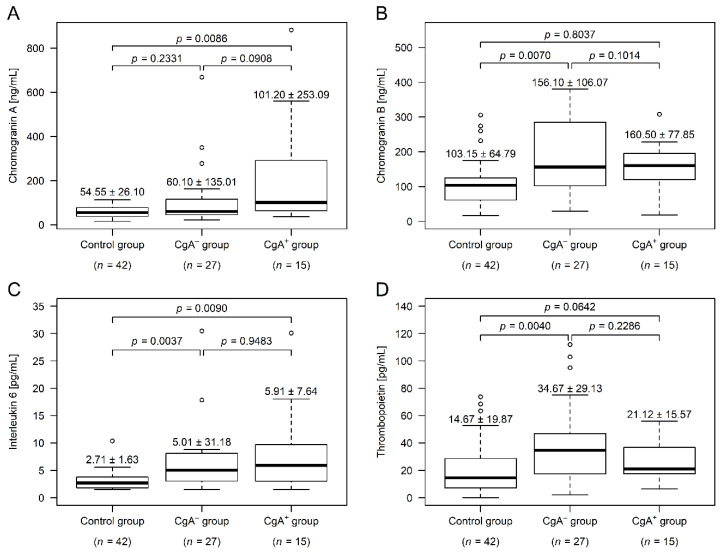
Chromogranin A, chromogranin B, interleukin-6, and thrombopoietin levels within the study groups (median ± standard deviation). Serum chromogranin A level (**A**) was significantly higher in patients with chromogranin A-positive neuroendocrine-cell differentiation within the colorectal adenocarcinoma (CgA^+^), while serum chromogranin B level (**B**) was higher in patients without chromogranin A-positive neuroendocrine-cell differentiation (CgA^−^). Plasma interleukin-6 (**C**) was elevated in both cancer groups, compared to those of healthy controls. In the CgA^−^ group there were four additional interleukin-6 outliers over 40 pg/mL (55.58, 77.19, 82.37, and 127.60 pg/mL), which have been cut off from the top of the figure for better visibility. The plasma thrombopoietin level (**D**) was higher within CgA^−^ patients, compared to both the other cancer group and the healthy controls.

**Figure 2 cancers-13-00067-f002:**
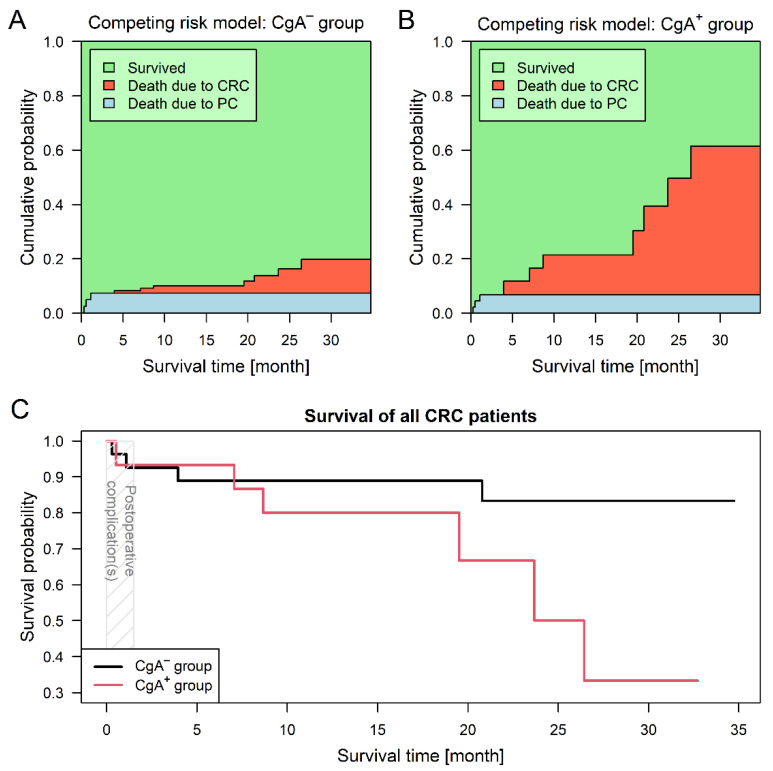
Survival analysis results of colorectal cancer (CRC) patients with (CgA^+^) and without (CgA^−^) chromogranin A-positive neuroendocrine-cell differentiation within the tumor. Three patients died due to postoperative complications (PC). Predictions based on the competing risk survival model (Aalen–Johansen estimator) suggest that patient survival was better within the CgA^−^ group (**A**), while patients within the CgA^+^ group (**B**) were much more exposed to tumor-related mortality (*p* = 0.0378). Combined curves (**C**) of the two study groups also suggested that CgA^+^ patients had a worse prognosis.

**Figure 3 cancers-13-00067-f003:**
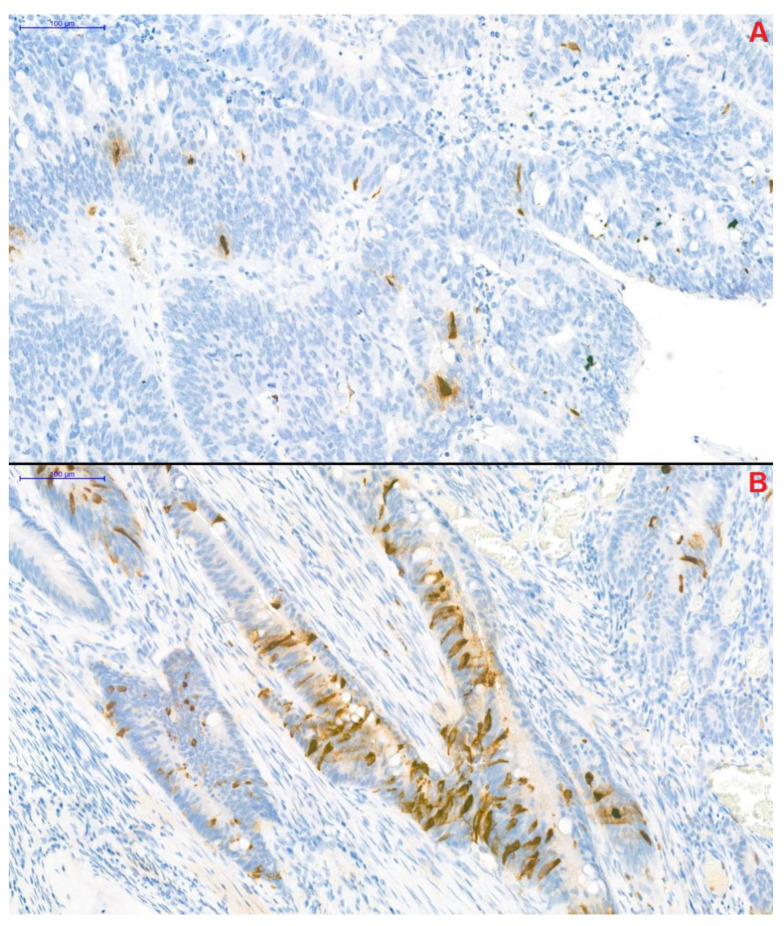
Chromogranin A-specific immunohistochemical staining of colorectal cancer samples (magnification: 20×). Two distinguishable pattern of the chromogranin A-positive neuroendocrine (CgA^+^) cells could have been observed within the samples. In the first, CgA^+^ cells were located individually in a scattered pattern (**A**), while in the remaining samples CgA^+^ cells formed small groups (**B**).

**Table 1 cancers-13-00067-t001:** Anamnestic data of colorectal cancer (CRC) patients, prior primary tumor removal (median ± standard deviation). Unit of frequency data is the number of observations.

Parameter	CgA^−^ Group (*n* = 27)	CgA^+^ Group (*n* = 15)	Uncorrected *p*-Value ^1^	FDR-Corrected *p*-Value
Age (years)	67.60 ± 7.90	69.00 ± 10.74		0.9113
Sex (male: female)	17: 10	9: 6		1.0000
Staging (AJCC [16])				
Stage I	7	0		
Stage II	9	6	0.0838	0.3831
Stage III	3	2		
Stage IV	8	7		
Side of CRC				
Left-sided	22	7	0.0352	0.2462
Right-sided	5	8		
Chemotherapy				
Adjuvant	6	3		
First-line	4	3		1.0000
Second-line	2	2		
Third-line	1	0		
Usage of biological therapy	5	2		1.0000
Comorbidities				
Diabetes	5	4		1.0000
Hypertension	20	10		1.0000
Major cardiovascular event prior CRC	4	6		0.3831
Thyroid diseases	2	1		1.0000
Drug treatment				
Aggregation inhibition	2	7	0.0055	0.0765
Statin	5	6		0.3831
Agents acting on the renin-angiotensin system	18	8		0.8947
Antacid therapy	7	6		0.8947

Notes: ^1^
*p*-values without false discovery rate (FDR) correction, where *p* < 0.1.

**Table 2 cancers-13-00067-t002:** Laboratory measurements of colorectal cancer patients prior to primary tumor removal (median ± standard deviation).

Parameter	CgA^−^ Group (*n* = 27)	CgA^+^ Group (*n* = 15)	Uncorrected *p*-Value ^1^	FDR-Corrected *p*-Value
Waist circumference (cm)	103.00 ± 11.11	94.50 ± 9.97		0.0222
Hip circumference (cm)	108.00 ± 7.88	101.50 ± 5.81		0.0099
Body mass index (kg/m^2^)	28.70 ± 3.98	27.00 ± 3.23		0.0465
White blood cell count (10^9^/L)	8.37 ± 7.02	8.26 ± 1.89		0.9113
Neutrophil count (10^9^/L)	5.46 ± 5.14	6.14 ± 1.72		0.3613
Eosinophil count (10^9^/L)	0.14 ± 1.68	0.14 ± 0.18		0.9903
Basophil count (10^9^/L)	0.05 ± 0.05	0.06 ± 0.03		0.9113
Monocyte count (10^9^/L)	0.54 ± 0.40	0.51 ± 0.20		0.9442
Lymphocyte count (10^9^/L)	1.67 ± 0.56	1.48 ± 0.43		0.0426
Red blood cell count (10^12^/L)	4.75 ± 0.41	4.52 ± 0.49		0.3118
Hemoglobin (g/L)	137.00 ± 20.51	118.00 ± 17.58		0.0099
Hematocrit (L/L)	0.41 ± 0.05	0.37 ± 0.04		0.0119
Mean corpuscular volume (fL)	85.50 ± 6.16	82.70 ± 8.96		0.2660
Mean corpuscular hemoglobin (pg)	28.10 ± 2.76	26.10 ± 4.09	0.0582	0.1212
Mean corpuscular hemoglobin concentration (g/L)	330.00 ± 16.04	316.00 ± 17.09		0.0426
Red blood cell distribution width (%)	13.20 ± 2.94	14.40 ± 5.20	0.0522	0.1186
Platelet count (10^9^/L)	292.00 ± 98.94	286.00 ± 128.38		0.3804
Lactate dehydrogenase (U/L)	202.00 ± 120.25	204.00 ± 669.73		0.9113
Estimated glomerular filtration rate $\left(\frac{\mathrm{mL}}{\min\cdot 1.73m^{2}}\right)$	90.60 ± 14.27	92.70 ± 25.19		0.9113
High-density cholesterol (mmol/L)	1.29 ± 0.36	1.36 ± 2.94		0.9113
Low-density cholesterol (mmol/L)	3.79 ± 0.93	2.88 ± 0.79		0.0119
High sensitivity C reactive protein (mg/L)	6.40 ± 50.46	7.10 ± 47.79		0.9113
Serum total protein (g/L)	74.00 ± 4.79	67.90 ± 4.68		0.0256
Serum albumin (g/L)	42.70 ± 5.02	38.10 ± 4.64		0.0222
Carcinoembryonic antigen (ng/mL)	3.70 ± 257.75	6.20 ± 958.29		0.7094
Carbohydrate antigen 19-9 (U/mL)	7.04 ± 1684	10.98 ± 12591		0.9055

Notes: ^1^
*p*-values without false discovery rate (FDR) correction, where *p* < 0.1.

**Table 3 cancers-13-00067-t003:** Laboratory measurements of colorectal cancer patients, at least six weeks after primary tumor removal (median ± standard deviation).

Parameter	CgA^−^ Group (*n* = 21)	CgA^+^ Group (*n* = 10)	Uncorrected *p*-Value ^1^	FDR-Corrected *p*-Value
Waist circumference (cm)	103.00 ± 9.19	93.00 ± 4.55		0.0010
Hip circumference (cm)	111.00 ± 8.25	103.50 ± 4.81		0.0082
Body mass index (kg/m^2^)	28.40 ± 4.15	25.50 ± 2.13		0.0046
White blood cell count (10^9^/L)	7.00 ± 1.80	7.37 ± 1.78		0.8506
Neutrophil count (10^9^/L)	4.32 ± 1.47	5.95 ± 1.74		0.8506
Eosinophil count (10^9^/L)	0.17 ± 0.11	0.12 ± 0.16		1.0000
Basophil count (10^9^/L)	0.05 ± 0.04	0.05 ± 0.03		0.8516
Monocyte count (10^9^/L)	0.45 ± 0.18	0.48 ± 0.11		0.8506
Lymphocyte count (10^9^/L)	1.61 ± 0.62	1.61 ± 0.48		0.8506
Red blood cell count (10^12^/L)	4.64 ± 0.50	4.75 ± 0.63		0.8506
Hemoglobin (g/L)	136.00 ± 18.52	126.50 ± 15.23	0.0346	0.1125
Hematocrit (L/L)	0.41 ± 0.05	0.40 ± 0.05		0.8506
Mean corpuscular volume (fL)	87.00 ± 5.78	83.00 ± 7.18	0.0151	0.0561
Mean corpuscular hemoglobin (pg)	29.80 ± 2.70	26.55 ± 2.95		0.0046
Mean corpuscular hemoglobin concentration (g/L)	330.00 ± 13.39	316.50 ± 10.78		0.0041
Red blood cell distribution width (%)	13.60 ± 2.41	16.50 ± 3.40		0.0041
Platelet count (10^9^/L)	290.00 ± 83.74	287.50 ± 128.95		1.0000
Lactate dehydrogenase (U/L)	182.00 ±57.81	195.00 ± 1063.21		0.7870
$\mathrm{Estimated}\;\mathrm{glomerular}\;\mathrm{filtration}\;\mathrm{rate}\;\left(\frac{\mathrm{mL}}{\min\cdot 1.73m^{2}}\right)$	84.80 ± 13.42	89.85 ± 15.37		0.6855
High density cholesterol (mmol/L)	1.43 ± 0.35	1.40 ± 0.30		0.8891
Low density cholesterol (mmol/L)	3.42 ± 0.82	3.49 ± 1.12		0.8506
High sensitivity C reactive protein (mg/L)	3.90 ± 10.90	2.15 ± 61.25		0.8506
Serum total protein (g/L)	73.40 ± 4.27	75.70 ± 5.18		0.8506
Serum albumin (g/L)	42.75 ± 4.48	41.10 ± 5.35		0.8506

Notes: ^1^
*p*-values without false discovery rate (FDR correction), where *p* < 0.1.

**Table 4 cancers-13-00067-t004:** Interleukin-6 and thrombopoietin measurements before and after primary tumor removal surgery (median ± standard deviation). A statistically marginal decrease was only observed in interleukin-6 levels in the case of patients without chromogranin A-positive neuroendocrine-cell differentiation (CgA^−^; *p* = 0.0995).

Group	Measurement	Interleukin-6 (pg/mL)	Thrombopoietin (pg/mL)
CgA^−^ group (*n* = 21)	Preoperative	4.83 ± 17.29	36.62 ± 28.63
	Postoperative	3.70 ± 3.35	30.10 ± 24.60
CgA^+^ group (*n* = 10)	Preoperative	5.01 ± 9.06	21.02 ± 12.51
	Postoperative	4.59 ± 15.13	28.15 ± 16.43

Notes: CgA^+^: patients with chromogranin A-positive neuroendocrine-cell differentiation within the colorectal adenocarcinoma.

## Data Availability

The datasets used and/or analysed during the current study are available from the corresponding author on reasonable request.

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
