# Peer review of "Histopathological Chromogranin A-Positivity Is Associated with Right-Sided Colorectal Cancers and Worse Prognosis"

_cancers, 2020, doi:10.3390/cancers13010067_

Round 1

Reviewer 1 Report

The manuscript „Histopathological Chromogranin A-Positivity is Associated with Right-Sided Colorectal Cancers and Worse Prognosis” by Zoltan Herold and colleagues describes a small study on colorectal cancer. Specifically, the authors analyze the chromogranin A-positive neuroendocrine-cell differentiation (CgA+) and group their patients according to this. They find that patients in the CgA+ group have worse survival and propose that a new subtype of CRC based on chromogranin A-positive neuroendocrine-cell differentiation.

Overall, this is a very solid manuscript that is easy to follow. I have two minor points that the authors should address:

  1. The study is based on a small number of tumor samples. Limitations of the study that arise when only 42 samples are analyzed should be pointed out.
  2. The plots in Figure 1 are unclear. Please define what the box plot indicates, what the whiskers show, and what the solid black line stands for. Furthermore, the numbers (mean +/- SD?) above the plot have to be defined. It is e.g. unclear why IL-6 in the CgA- group is 17.80, while it is 7.83 in the CgA+ group (the box plots indicate otherwise). Please check and add a better description to the figure legends.

Author Response

Dear Reviewer,

Thank you for your positive feedback on our article. Here we provide answers to the questions you have raised:

The study is based on a small number of tumor samples. Limitations of the study that arise when only 42 samples are analyzed should be pointed out.

Thank you for bringing our attention to include a section about the limitations of our study. A short paragraph has been inserted at the end of Discussion to point out limitation, including sample size, of the study.

The plots in Figure 1 are unclear. Please define what the box plot indicates, what the whiskers show, and what the solid black line stands for. Furthermore, the numbers (mean +/- SD?) above the plot have to be defined. It is e.g. unclear why IL-6 in the CgA- group is 17.80, while it is 7.83 in the CgA+ group (the box plots indicate otherwise). Please check and add a better description to the figure legends.

Thank you for the suggestion. Data had been described originally with mean ± SD, throughout the manuscript. In several cases, like IL-6, data were left skewed, which was caused by outlieres. The statistical software R do not draw boxplots by default with the mean, but with the median. To preserve the uniform impression of the manuscript, we replaced means with medians in both the tables and the figures. Furthermore, a new paragraph, detailing how boxplots are drawn, had been added to the end of Methods.

Yours sincerely,

                                                       Zoltan Herold

                                                  Semmelweis University

                                  Department of Internal Medicine and Hematology

Reviewer 2 Report

The Authors performed a prospective study to analyze the levels of Chromogranin A in patients with colorectal cancer. They found that high levels of Chromogranin A were correlated with worse clinical outcome, independently by the stage of the tumor.

Chromogranin A levels were also correlated with paraneoplastic thrombocytosis, even if the mechanisms is not clear.

Based on these observations the Authors propose a new subtype of CRC,
which can be characterized by chromogranin A-positive neuroendocrine-cell
differentiation.

The study is well done and opens new horizons , or at least supports new ideas about the pathophysiology of some colorectal cancers.

There are some major points I would like to underline about this study.

1)Chromogranin A was detected by immunoghystochemistry. This test is  a subjective test, which can give only a qualitative assessment. Was this a blind examen? Were  the pathologists unaware of the clinical outcomes of the patients, as well as of the hematologic conditions of the patents?

2)How did the Authors describe, and which parameters they used to report  the results of the immunohistochemistry?

3)In this context, RNA polymerasis may give more objective information allowing a quantitative evaluation..

4)Chromogranin A, due to its primary expression throughout the neuroendocrine system, is a widely accepted biomarker for the assessment of neuro-endocrine tumors. It has been traditionally used in the management of patients with tumors of gastro-enteropancreatic origin. The Granins comprise a family of proteins whose most well known members are chromogranin A (CgA), chromogranin B (CgB) and secretogranin II.  Other proteins that are also included in the granin family are secretogranin III, the HISL-19 antigen (secretogranin IV), the neuroendocrine secretory protein 7B2 (secretogranin V), NESP55 (secretogranin VI) and nerve growth factor-inducible protein VGF (secretogranin VII). The overall sensitivity of CgA in the diagnosis of neuroendocrine tumors is around 60%-80% and depends on the primary site, on the degree of differentiation and on the status of the disease This marker has a low sensitivity regarding its use in distinguishing the different types of tumors.. It should be noted also, that the specificity and sensitivity of the assay for CgA measurement differ between the available commercial kits.

The possibility that the results of thiswstudy can be conditioned by the used  commercial Kit is an important matter to  be added in the discussion

5)T A prognostic correlation has been found between reduced levels of  levels of Chromogranin A after therapy and better clinical outcome. Those results have been confirmed in a relevant analysis of the phase III RADIANT-2 clinical trial, where it was shown that early decrease of CgA levels by Everolimus can be used as a surrogate marker of  cancer free survival in this setting.

How the Authors explain the increase of Chromogranin A after surgery in patients CgA positive?

6) Chromogranin A levels are increased in patients with neuro-endocrine tumors. Even  patients with non-neuroendocrine tumors may also have abnormal levels of circulating CgA. For example, increased levels of immunoreactive CgA have been observed in a subpopulation of patients with non-small cell lung cancer lacking neuroendocrine cells in tumor tissues and in cancer patients treated with proton pump inhibitors, a class of drugs commonly used to treat acid peptic disorders Elevated serum levels of CgA have been observed in patients with renal failure, heart failure, hypertension, rheumatoid arthritis, atrophic gastritis, inflammatory bowel disease, sepsis and other inflammatory diseases 

The Authors cannot exclude that high serum levels of Chromogranin A were not an index of inflammation related to the cancer. Indirectly, this possible correlation between high levels of Chromogranin A and inflammation is supported by increased levels of Interleukin 6, which has a well known inflammatory action.

From this matter, we might hypothesize that the positivyt at pathology for Chromogranin A was an index of inflammation per se.

In this context measurement of other inflammatory indices (Protein C Reactive, TNF alfa, IL 1) might provide useful information.

7) Interestingly, recent studies have shown that CgA  and the N-terminal fragment CgA inhibit angiogenesis, whereas the fragment CgA stimulates angiogenesis in various angiogenesis assays It is possible  that the site responsible for the anti-tumor effects of full-length CgA is located in the C-terminal region. ChromograninA fragmentation, losing the C-terminal , may determine increased permeability of endothelial cells and inflammation.

Thus, the matter is more complex and it is difficult to understand if positivity for Chromogranin A is an index of a new pathophysiology for colon cancer or just an index for the inflammation associated with the cancer.

8) Systemic levels of CgA for GEP-NETs appear to be higher for well vs poorly differentiated tumors, functioning vs non-functioning, metastatic vs locoregional disease. It could be interesting to correlate Chromogranin histochemistry and serum levels with the differentiation of colon cancer cells. The possibility that CgA levels are correlated with more differentiated tumor might  provide  an indication for the presence of a strong component of neuroendocrine differentiation within an adenocarcinoma.

MINOR POINTS

In general material and methods are described after the introduction and before the results.

Final comment

Interesting study. The major questions is related to the meaning of increased CgA levels: inflammation related or an index of the presence of neuroendocrine cells within a  colon adenocarcinoma?

As all interesting studies, the paper give more questions than answers.

Author Response

Dear Reviewer,

Thank you for your positive feedback on our article. Our answers for your critical comments and questions are below:

1. Chromogranin A was detected by immunohistochemistry. This test is a subjective test, which can give only a qualitative assessment. Was this a blind examen? Were the pathologists unaware of the clinical outcomes of the patients, as well as of the hematologic conditions of the patents?

Pathologist did know only the location and staging of the tumor but was unaware of any of other clinical details or survival data of patients, including those of CgA-, CgB-, interleukin-6-, thrombopoietin levels and other laboratory measurements. A sentence has been added to Methods detailing the blindness of examination.

2. How did the Authors describe, and which parameters they used to report the results of the immunohistochemistry?

Results of immunohistochemistry was reported similar to those of described by Gulubova and Vlaykova (Ref. no. 12 within the manuscript): “In brief, under low power magnification of the microscope (x100), the tumor tissue was assessed as being positive or negative (for endocrine cells).” Text within the manuscript had been updated to the following: “CgA-positivity was defined as previously described by Gulubova et al. [12]: Adenocarcinoma samples, in which anti-CgA-stained cells infiltrated beyond the muscularis mucosae were defined as CgA-positive.”

3. In this context, RNA polymerasis may give more objective information allowing a quantitative evaluation.

Thank you for the suggestion. We agree with the Reviewer that information on chromogranin specific RNA quantities would describe the neuroendocrine-like status of the tumor more precisely. Unfortunately, the original design of the study did not include preparing and storing RNA-fixed samples from the tumors, which would be essential to perform such an analysis. RNA analysis on remaining samples is not feasible due to the degradation of RNA caused by frozen storage.

4. Chromogranin A, due to its primary expression throughout the neuroendocrine system, is a widely accepted biomarker for the assessment of neuro-endocrine tumors. It has been traditionally used in the management of patients with tumors of gastro-enteropancreatic origin. The Granins comprise a family of proteins whose most well-known members are chromogranin A (CgA), chromogranin B (CgB) and secretogranin II. Other proteins that are also included in the granin family are secretogranin III, the HISL-19 antigen (secretogranin IV), the neuroendocrine secretory protein 7B2 (secretogranin V), NESP55 (secretogranin VI) and nerve growth factor-inducible protein VGF (secretogranin VII). The overall sensitivity of CgA in the diagnosis of neuroendocrine tumors is around 60%-80% and depends on the primary site, on the degree of differentiation and on the status of the disease This marker has a low sensitivity regarding its use in distinguishing the different types of tumors. It should be noted also that the specificity and sensitivity of the assay for CgA measurement differ between the available commercial kits.

The possibility that the results of this study can be conditioned by the used commercial Kit is an important matter to be added in the discussion

Thank you for the suggestion. It was added as one of the limitations of study (please see ‘Limitations of the study’, at the end of Discussion).

5. A prognostic correlation has been found between reduced levels of Chromogranin A after therapy and better clinical outcome. Those results have been confirmed in a relevant analysis of the phase III RADIANT-2 clinical trial, where it was shown that early decrease of CgA levels by Everolimus can be used as a surrogate marker of cancer free survival in this setting.

How the Authors explain the increase of Chromogranin A after surgery in patients CgA positive?

Unresectable pNET patients in the RADIANT-2 study (Pavel et al., Lancet 2011) were treated with octreotide + everolimus or octreotide + placebo, from which octreotide is known to decrease circulating CgA levels (Moattari & Vinik, J Clin Endocrinol Metab 1989). In the current pilot study, CRC patients did not receive any oncological treatment that may decrease chromogranin levels, only the, currently used standardized, “CRC-specific” chemotherapy protocols were used.

Postoperative CgA level was not presented within the original version of the manuscript. Patients are usually treated with various medications for prophylaxis, and in several cases antacids are widely used after the diagnosis of the tumor as well. Due to patient safety, no therapy was stopped/paused prior any measurements, which resulted in a very small number of CgA measurement sample size to compare with preoperative results. 13 of the 31 patients had reliable CgA follow-up measurements which was not affected by the use of antacids. Preoperative CgA level of those 13 patients was 77.7 ± 129.68 ng/mL, while postoperative was 61.4 ± 92.61 ng/mL. Both traditional and permutation-based paired t-test showed no significant difference between the two measurements (p = 0.2335, permutation-based: p = 0.2224). However, due to the very low number of samples, results of these tests have low statistical power. It is the reason behind we did not include these data, and the following sentence was included only within the original version of the manuscript: “Follow-up measurement of CgA could not be adequately evaluated due to the low number of cases and the high number of antacid usage.”

6. Chromogranin A levels are increased in patients with neuro-endocrine tumors. Even patients with non-neuroendocrine tumors may also have abnormal levels of circulating CgA. For example, increased levels of immunoreactive CgA have been observed in a subpopulation of patients with non-small cell lung cancer lacking neuroendocrine cells in tumor tissues and in cancer patients treated with proton pump inhibitors, a class of drugs commonly used to treat acid peptic disorders. Elevated serum levels of CgA have been observed in patients with renal failure, heart failure, hypertension, rheumatoid arthritis, atrophic gastritis, inflammatory bowel disease, sepsis and other inflammatory diseases 

The Authors cannot exclude that high serum levels of Chromogranin A were not an index of inflammation related to the cancer. Indirectly, this possible correlation between high levels of Chromogranin A and inflammation is supported by increased levels of Interleukin 6, which has a well-known inflammatory action.

From this matter, we might hypothesize that the positivity at pathology for Chromogranin A was an index of inflammation per se.

In this context measurement of other inflammatory indices (Protein C Reactive, TNF alfa, IL 1) might provide useful information.

Thank you for the suggestion to further investigate chromogranins and inflammatory markers. No difference was found between the two tumor group within inflammatory markers, such as high sensitivity C reactive protein (hsCRP), interleukin-6 and white blood cell counts (please see Table 2 and Figure 1). Correlation results and a section about inflammation had been inserted to Results and Discussion, respectively.

Exclusion criteria for the study were similar in many respects to diseases and/or symptoms that would artificially increase CgA levels, except the usage of antacids, as detailed in Methods. Results were calculated without patients using antacids as well; and the same could have been observed in this subset as well.

7. Interestingly, recent studies have shown that CgA and the N-terminal fragment CgA inhibit angiogenesis, whereas the C-terminal fragment CgA stimulates angiogenesis in various angiogenesis assays. It is possible that the site responsible for the anti-tumor effects of full-length CgA is located in the C-terminal region. Chromogranin A fragmentation, losing the C-terminal, may determine increased permeability of endothelial cells and inflammation.

Thus, the matter is more complex, and it is difficult to understand if positivity for Chromogranin A is an index of a new pathophysiology for colon cancer or just an index for the inflammation associated with the cancer.

Markers of inflammation (white blood cell count, hsCRP and interleukin-6) correlated only with circulating CgB levels, while in the case CgA no significant correlation was found. Please see our answer to question no.6.

In a recent article (Curnis et al., Oncotarget, 2016) the cleavage product CgA410-439 was found to have angiogenic and anti-tumor effects, which can be inhibited with neutralization with antibodies against this specific region. The results suggest that CgA, but mainly its C-terminal residues would have a role in these processes as well. We couldn’t agree more that the underlying mechanisms could be present in colorectal cancer as well, and these should be investigated in a model system. Unfortunately, we did not have the opportunity to test such hypothesis based on our clinical observations.

8. Systemic levels of CgA for GEP-NETs appear to be higher for well vs poorly differentiated tumors, functioning vs non-functioning, metastatic vs locoregional disease. It could be interesting to correlate Chromogranin histochemistry and serum levels with the differentiation of colon cancer cells. The possibility that CgA levels are correlated with more differentiated tumor might provide an indication for the presence of a strong component of neuroendocrine differentiation within an adenocarcinoma.

Subgrouping patients within the two tumor groups were not advisable due to the small sample size. We agree with the Reviewer that this is one of the questions that has to be answered in future investigations. The following CgA level were observed in our data (where n ≥ 5; median ± SD):

  • CgA group
    • Stage 1: 53.10 ± 42.57 (n = 7)
    • Stage 2: 68.96 ± 112.20 (n = 9)
    • Stage 4: 108.55 ± 209.00 (n = 8)
  • CgA+ group
    • Stage 2: 115.00 ± 215.69 (n = 6)
    • Stage 4: 147.70 ± 311.70 (n = 7)

Limitations of the study had been updated with this question as well.

9. In general, material and methods are described after the introduction and before the results.

The reason Methods have been placed after Discussion was as per ‘Author guidelines’ of the journal.

Yours sincerely,

                                                       Zoltan Herold

                                                  Semmelweis University

                                  Department of Internal Medicine and Hematology

Round 2

Reviewer 2 Report

The Authors have done appropriate corrections.

Overall, this is an interesting study which inroduces innovative hypothesis about the etiology of a selected group of patients with colorectal cancer.

Still, the basic question remains :  are high levels of Chromogranin A indicative of a different etiopathogenetic process? Any definitive answer requires identification of the neuroendocrine cells responsible for the increased production/release of Chromogranin A.

The possibility that Chromogranin A increased level is another epiphenomenon of a tumor-asscoiated inflammation remains.

The paper opens a new field of research in colorectal cancer etiology and possible future treatments.

I recommend to publish the paper.